# Curcumin promotes AApoAII amyloidosis and peroxisome proliferation in mice by activating the PPARα signaling pathway

Jian Dai[1,2]*, Ying Li[3], Fuyuki Kametani[4], Xiaoran Cui[5], Yuichi Igarashi[5], Jia Huo[6], Hiroki Miyahara[1,7], Masayuki Mori[1,7], Keiichi Higuchi[1,7]

[1]Department of Neuro-health Innovation, Institute for Biomedical Sciences, Interdisciplinary Cluster for Cutting Edge Research, Shinshu University, Matsumoto, Japan; [2]Department of Pathology, the Xiehe Hospital of Tangshan, Tangshan, China; [3]Aging Biology, Department of Biomedical Engineering, Graduate School of Medicine, Science and Technology Shinshu University, Matsumoto, Japan; [4]Department of Dementia and Higher Brain Function, Tokyo Metropolitan Institute of Medical Science, Tokyo, Japan; [5]Department of Aging Biology, Institute of Pathogenesis and Disease Prevention, Shinshu University Graduate School of Medicine, Matsumoto, Japan; [6]Department of Orthopedic Surgery, the Third Hospital of Hebei Medical University, Shijiazhuang, China; [7]Department of Aging Biology, Shinshu University School of Medicine, Matsumoto, Japan

*For correspondence:
daijian3@shinshu-u.ac.jp

Competing interests: The authors declare that no competing interests exist.

**Abstract** Curcumin is a polyphenol compound that exhibits multiple physiological activities. To elucidate the mechanisms by which curcumin affects systemic amyloidosis, we investigated amyloid deposition and molecular changes in a mouse model of amyloid apolipoprotein A-II (AApoAII) amyloidosis, in which mice were fed a curcumin-supplemented diet. Curcumin supplementation for 12 weeks significantly increased AApoAII amyloid deposition relative to controls, especially in the liver and spleen. Liver weights and plasma ApoA-II and high-density lipoprotein concentrations were significantly elevated in curcumin-supplemented groups. RNA-sequence analysis revealed that curcumin intake affected hepatic lipid metabolism via the peroxisome proliferator-activated receptor (PPAR) pathway, especially PPARα activation, resulting in increased *Apoa2* mRNA expression. The increase in liver weights was due to activation of PPARα and peroxisome proliferation. Taken together, these results demonstrate that curcumin is a PPARα activator and may affect expression levels of proteins involved in amyloid deposition to influence amyloidosis and metabolism in a complex manner.

## Introduction

Amyloidosis is a group of diseases characterized by abnormal aggregation of proteins to form amyloid fibrils, and subsequent deposition in various tissues and organs, which can lead to severe functional failures. More than 30 amyloid proteins have been identified; some result in localized tissue deposits, such as Aβ in Alzheimer's disease and α-synuclein (αSyn) in Parkinson's disease that deposit in the brain, while others result in systemic amyloidosis and are widely deposited in various tissues and organs, such as ALλ and ALκ in immunoglobulin light chain amyloidosis and transthyretin in ATTR amyloidosis (*Benson et al., 2018*). In general, when amyloid proteins are exposed to certain conditions that affect protein homeostasis (e.g., overexpression, gene mutation, and enzyme cleavage), they may undergo structural changes into stable structures that are rich in β-sheets, and which promote subsequent aggregation to form oligomers, protofibrils, and amyloid fibrils (*Merlini and*

*Bellotti, 2003*; *Knowles et al., 2014*). Because the formation of amyloid fibrils is nearly irreversible, maintaining proteostasis and inhibiting amyloid aggregation present a challenge for development of an effective treatment.

Some natural phenolic compounds extracted from plants exhibit certain anti-amyloid activity in vitro and in vivo (*Stefani and Rigacci, 2013*). Curcumin, a polyphenol compound, is extracted from the rhizome of *Curcuma longa* and has a long history of use in traditional medicines in some countries in Asia. In in vitro experiments, curcumin has been shown to suppress the aggregation and cytotoxicity of Aβ, αSyn, islet amyloid precursor protein, ATTR, and prion protein (*Stefani and Rigacci, 2013*). In 2001, the first evidence of the efficacy of curcumin against Aβ amyloidosis in a transgenic model mice was reported (*Lim et al., 2001*). Curcumin was found to suppress amyloid deposition in a mouse model of Alzheimer's disease and improve memory function. It was subsequently demonstrated that the amount of amyloid present in TTR- and tau-transgenic mice was reduced by curcumin supplementation (*Ferreira et al., 2013*; *Ferreira et al., 2016*; *Ma Q-L et al., 2013*). Due to the strong affinity of curcumin for the amyloid structure, it is believed that curcumin inhibits the formation of amyloid fibrils by binding to amyloid protein monomers or aggregates (*Stefani and Rigacci, 2013*; *Lim et al., 2001*; *Ahmad et al., 2017*; *Hafner-Bratkovic et al., 2008*). This curcumin-protein complex exhibits better stability and reduces the tendency to aggregate. However, another mechanism that has been proposed suggests that curcumin inhibits Aβ production by downregulating the expression of amyloid-beta precursor protein (APP) or beta-site APP cleaving enzyme 1 (BACE1) in vitro (*Song et al., 2020*; *Zheng et al., 2017*). Unfortunately, there are few reports that suggest that curcumin will provide clinical benefit in patients with Alzheimer's disease or AL amyloidosis (*Stefani and Rigacci, 2013*; *Golombick et al., 2015*; *Small et al., 2018*). In fact, it is unclear how curcumin inhibits amyloid deposition in vivo.

Curcumin is a compound with multiple physiological activities, which include anti-oxidation, anti-inflammatory, anti-cancer, lipid metabolism regulation, and anti-amyloid properties. However, a link between the various physiological activities has not been completely established (*Liczbiński et al., 2020*). Curcumin has been found to exert an influence on multiple signaling pathways (*Shishodia, 2013*). In 2003, curcumin was first shown to inhibit rat hepatic stellate cell growth by activation of peroxisome proliferator-activated receptor γ (PPARγ), suggesting that curcumin might have an effect on the PPAR signaling pathway (*Xu et al., 2003*). In mammals, the PPAR subfamily (PPARs) is a group of nuclear receptor proteins, for example transcription factors, and consists of three members, namely PPAR-α, PPAR-β/δ, and PPAR-γ, that play essential roles in the regulation of metabolic homeostasis, glucose and energy metabolism, cellular differentiation, inflammation, and ROS metabolism (*Pyper et al., 2010*; *Monsalve et al., 2013*; *Lamichane et al., 2018*; *Harmon et al., 2011*). The functions of the three PPAR subtypes are different. PPAR-α regulates fatty acid transport and oxidative decomposition in the liver and muscle in response to energy metabolism levels. PPARγ mainly regulates fatty acid synthesis and fat accumulation in adipose tissue, as well as differentiation of adipose cells and macrophages. PPARβ/δ plays an important role in lipid catabolism, energy homeostasis, and cell differentiation, but the mechanism and network of action are not completely clear (*Harmon et al., 2011*; *Magadum and Engel, 2018*). Among the three isotypes, the relationship between curcumin and PPARγ is the most extensively studied, while information on α and β/δ remains scarce.

In this study, we sought to determine whether curcumin affects the amyloid deposition process besides directly binding to amyloid proteins, and identify a link between curcumin's anti-amyloid activity and its various other biological activities. In our previous study, we found that antioxidants (tempol and apocynin) can effectively reduce AApoAII amyloid deposition (*Dai et al., 2019*). It is therefore possible that the anti-oxidative effects of curcumin also play an important role in amyloid formation. We examined the effects of curcumin supplementation in a mouse model of AApoAII amyloidosis, in which mice were induced to develop systemic amyloidosis (*Higuchi et al., 1995*). In contrast to expectations, our results showed that curcumin significantly promoted AApoAII amyloid deposition by activating the PPAR signaling pathway. Moreover, our results suggest that activation of PPARα plays a major role in the amyloid formation process.

## Results

### Degree of AApoAII amyloid deposition and liver weights were significantly increased after supplementation with curcumin

Two-month-old female R1.P1-$Apoa2^c$ mice were randomly divided into four groups: the control (Con) and curcumin (Cur) groups are non-amyloid-induced groups and fed a common diet or 2% w/w curcumin diet, respectively. The other two groups were injected with 1 µg AApoAII amyloid fibrils into the tail vein to induce amyloidosis, and were fed a common diet (A-NT group) or 2% curcumin diet (A-Cur group) (see experimental design in *Figure 1—figure supplement 1*, body weight and food intake in *Figure 1—figure supplement 2*). After 8 and 12 weeks, we evaluated the effect of curcumin intake on amyloid deposition. Unlike previous studies, the degree of amyloid deposition in the A-Cur group was significantly increased compared to the A-NT group at both 8 and 12 weeks, especially in the liver and spleen (*Figure 1a–d*, *Figure 1—figure supplement 3*). This is the first evidence that curcumin promotes amyloid deposition in vivo and suggests that curcumin can affect the amyloid deposition process via a complex mechanism, not just via binding with amyloid protein monomers or aggregates. In addition, no amyloid deposition was observed in the Cur group, suggesting that curcumin does result in the pathogenesis of amyloidosis without induction by amyloid fibril injection. These results suggest that curcumin accelerates amyloid deposition but does not cause structural changes in amyloidogenic proteins.

On the other hand, we noticed that the livers of mice were larger and heavier in those mice that received dietary supplementation with curcumin than in those without supplementation (*Figure 1e and f*, *Supplementary file 1*). In the subsequent histological observation, it was found that the mice in the curcumin-supplemented groups had hepatocyte hypertrophy and abnormal changes in some hepatocyte nuclei, but no tumors or abnormal organisms were observed (*Figure 1—figure supplement 4*). This is somewhat confusing, as it has been demonstrated in many high-fat diet experiments that curcumin effectively reduces liver lipid deposition and liver weight (*Um et al., 2013*; *Maithilikarpagaselvi et al., 2016*). To exclude the effects of hepatocyte damage or inflammation, plasma levels of aspartate aminotransferase (AST) and alanine transaminase (ALT) and several inflammatory marker cytokines in the liver were detected (*Figure 1—figure supplements 5* and *6*). However, we found no support that dietary supplementation with curcumin causes inflammation or cell injury in the liver. To confirm whether different doses will affect the experimental results, we repeated the experiment with a lower dose (0.5% w/w) of curcumin diet and obtained similar changes in the degree of amyloid deposition and liver weight (*Figure 1—figure supplement 7*).

### Curcumin elevated levels of ApoA-II protein and affected lipid metabolism in mice

It has been reported that overexpression of amyloid protein or precursor protein is one of the most important factors contributing to the pathogenesis of amyloidosis, and results in increased amyloid deposition in transgenic model mice (*Merlini and Bellotti, 2003*; *Knowles et al., 2014*; *Calhoun et al., 1999*). Apolipoprotein A-I (ApoA-I) and apolipoprotein A-II (ApoA-II) are the major proteins comprising HDL (high-density lipoprotein) particles, and curcumin has been shown to increase HDL levels in some studies of lipid metabolism (*Yang et al., 2014*; *Panahi et al., 2018*). We previously demonstrated that overexpression of ApoA-II can significantly aggravate amyloid deposition in $Apoa2^c$ transgenic mice (*Ge et al., 2007*). Moreover, the co-deposited proteins in AApoAII amyloidosis may also affect the degree of amyloid deposition (*Miyahara et al., 2018*). We hypothesized that the increase in amyloid deposition in the curcumin diet group was due to upregulation of ApoA-II or other co-deposition proteins. Plasma levels of ApoA-II and the major proteins that were co-deposited with AApoA-II amyloids (ApoA-I and ApoE) were measured by western immunoblot. In line with our expectations, significantly higher plasma levels of ApoA-II at 12 weeks were observed for the Cur group compared with the Con group (*Figure 2a*). Following the increase in amyloid deposition, the plasma levels of ApoA-II will gradually decrease upon tissue deposition (*Miyahara et al., 2018*). As shown in *Figure 2a*, plasma levels of ApoA-II in the A-Cur group decreased significantly compared with those in the Cur group, while the decline in the A-NT group was mild compared with that in the Con group. These different degrees of decline resulted in a

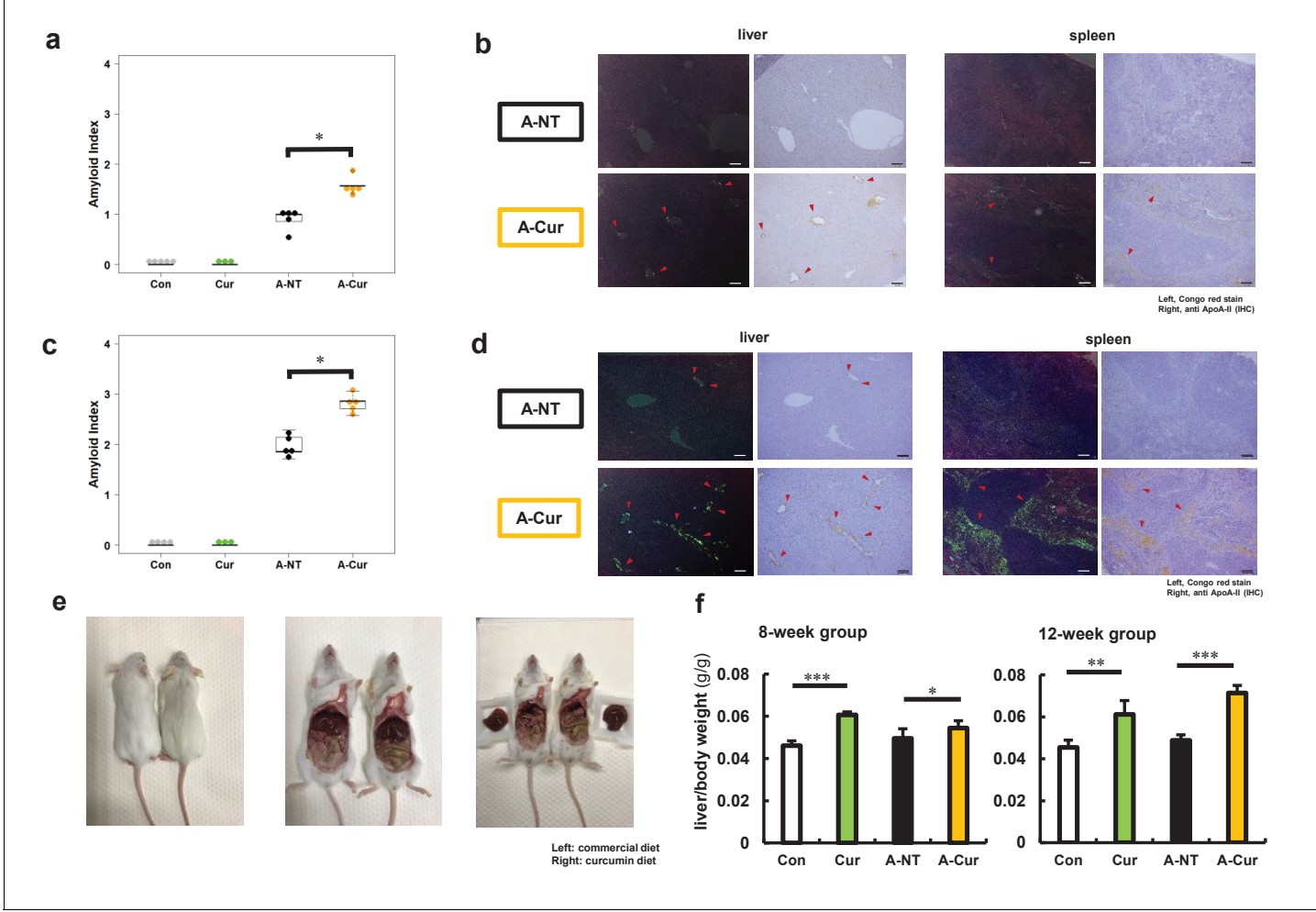

**Figure 1.** Degree of AApoAII amyloid deposition and liver weights. (**a**) Amyloid index (AI) in the 8-week group. (**b**) Representative Congo red and IHC images of AApoAII amyloid deposition in the 8-week group. Amyloid deposits (red arrows) were identified by green birefringence in Congo red-stained sections using polarizing light microscopy. Each scale bar indicates 100 μm. (**c**) Amyloid index in the 12-week group. (**d**) Representative Congo red and IHC images of amyloid deposition in the 12-week group. (**e**) Mice in curcumin-supplemented group have larger livers and less adipose tissue in the abdominal cavity than mice without curcumin supplementation (left: commercial diet; right: curcumin diet). (**f**) Ratio of liver weight/body weight in all groups. Each dot represents an individual mouse (**a**, **c**). Data are mean ± SD (**f**). N = 3–5. The Kruskal–Wallis test with the Steel–Dwass test was used for the amyloid index, and the Tukey–Kramer method was used for multiple comparisons of liver weights; *p < 0.05, **p < 0.01, ***p < 0.001.
The online version of this article includes the following figure supplement(s) for figure 1:

**Figure supplement 1.** Experimental design.
**Figure supplement 2.** Weekly body weight and food intake measurements.
**Figure supplement 3.** Amyloid score in various organs.
**Figure supplement 4.** Hepatocyte hypertrophy in curcumin intake groups.
**Figure supplement 5.** No differences in AST and ALT plasma concentrations after curcumin intake.
**Figure supplement 6.** Inflammation-related gene expression in the liver in the 12-week group.
**Figure supplement 7.** Degree of AApoAII amyloid deposition and liver weights were significantly increased after supplementation with low dose curcumin diet (0.5% w/w).

narrowing difference between the A-NT and A-Cur groups. ApoA-I plasma levels showed a slight increase with curcumin supplementation, but were not significant (*Figure 2b*). Consistent with previous results (*Dai et al., 2019*), ApoE, which is the most abundant co-deposited protein in AApoAII amyloidosis, was significantly increased in the amyloid-induced groups, but was not significantly affected by curcumin (*Figure 2c*).

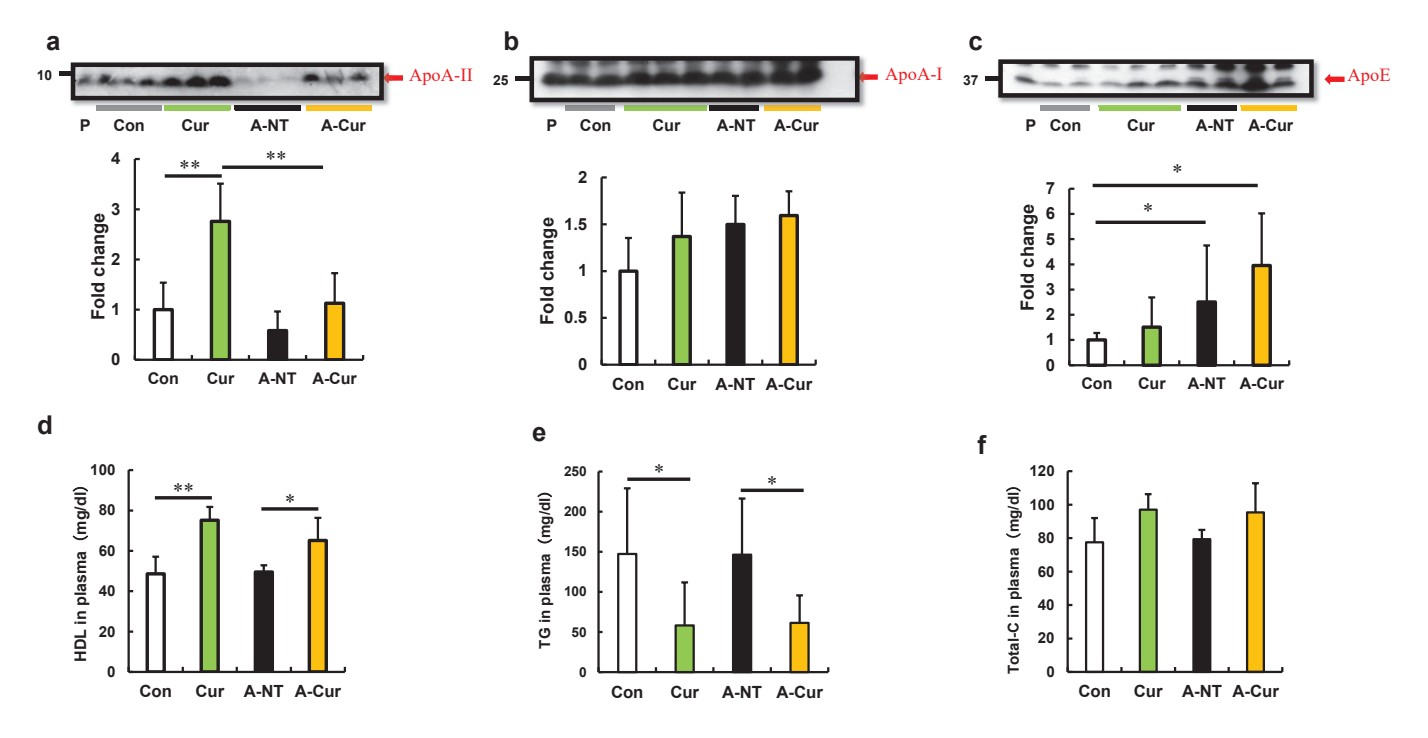

**Figure 2.** Curcumin elevated ApoA-II levels and affected lipid metabolism in mice after 12 weeks. (**a–c**) Plasma concentrations of ApoA-II, ApoA-I, and ApoE were determined by densitometry of Western immunoblot after SDS-PAGE. Representative results of western blot are shown above the figures. Histograms show fold changes relative to the Con group and represent the means ± SD. P indicates the pooled plasma of female R1.P1-Apoa2c mice at 2 months of age (N = 4) that did not have AApoAII amyloid deposits, as the positive control. (**d–f**) Plasma concentrations of HDL cholesterol, triglycerides (TG), and total cholesterol (Total-C) were determined using quantitative assay kits. N = 3–5. The Tukey–Kramer method was used for multiple comparisons; *p < 0.05. **p < 0.05.

The online version of this article includes the following figure supplement(s) for figure 2:

**Figure supplement 1.** ApoA-II, ApoA-I, and ApoE plasma levels after supplementation with 2% w/w curcumin diet for 8 weeks.

Because ApoA-II, ApoA-I, and ApoE are all constituent proteins of HDL and are related to lipid metabolism, we evaluated the change in lipid metabolism by detecting plasma levels of total cholesterol, HDL cholesterol, and triglycerides. Results suggest that curcumin supplementation increased HDL cholesterol levels, but reduced triglycerides levels (*Figure 2e–g*), which is consistent with previous experiments of curcumin supplementation. Similar results were also shown in lipid metabolism studies, in which hypolipidemic agents, namely fibrates, were used as PPARα agonists (*van der Hoogt et al., 2007*; *Staels et al., 1998*).

Further, immunoblotting results of ApoA-I and ApoE in the 8-week group are consistent with those of the 12-week group, but the change in ApoA-II protein did not show a significant difference. These results indicate that the increase in ApoAII levels may have undergone a slow process to adapt to the changes in lipid metabolism (*Figure 2—figure supplement 1*).

## RNA sequence analysis showed that curcumin regulates many lipid metabolism-related genes via the PPAR signaling pathway, especially by activating PPARα in the liver

Although it has been suggested that curcumin inhibits amyloid fibril formation by binding to amyloid proteins and maintaining protein homeostasis, our results showed that curcumin may also affect the degree of amyloid deposition by other means. Several physiological activities of curcumin are related to the activation of PPARγ, including decreased insulin resistance, anti-inflammatory, and anti-cancer activities (*Liczbiński et al., 2020*; *Shishodia, 2013*; *Janani and Ranjitha Kumari, 2015*; *Youssef and Badr, 2011*). However, ApoAII expression is thought to be upregulated upon activation of PPARα to enhance the delivery of lipids from the periphery tissue to the liver (*Liu et al., 2015*;

Shah et al., 2010). To identify possible signaling pathways or target proteins that respond to curcumin, we performed a comprehensive analysis of mRNA transcription in the liver using the RNA sequence method. As shown in the Venn diagram (*Figure 3a*, *Figure 3—source datas 1* and *2*), 75 genes are changed in mRNA expression by curcumin supplementation regardless of induction of amyloidosis (*Figure 3—source data 3*). The enrichment pathway analysis based on the KEGG database suggested that differentially expressed genes (DEGs) are mainly distributed in lipid metabolism-related signaling pathways centered on the PPAR pathway (*Figure 3b*, *Figure 3—source datas 4* and *5*). Analysis of the non-amyloid induced groups separately showed that 98 genes were upregulated and 93 genes were downregulated in the Cur group compared with the Con group. Among the 15 genes that were most significantly upregulated, more than 2/3 were related to fatty acid transport and fatty acid oxidation (*Figure 3c*). Even considering the possible interference of AApoAII amyloidosis on gene expression in amyloid-induced groups, most of these genes had still been promoted in the A-Cur group (*Figure 3d*).

As a transcription factor, PPARα is a major regulator of lipid metabolism in the liver. When the body is in an energy-deprived state, activated PPARα can mobilize fatty acids to the liver and promote fatty acid β oxidation to produce energy by upregulation of genes involved in fatty acid transport, fatty acid binding, and peroxisomal and mitochondrial fatty acid β-oxidation (*Pyper et al., 2010*; *Monsalve et al., 2013*; *Lamichane et al., 2018*). During lipid metabolism, the physiological function of PPARγ mainly involves synthesis and elongation of fatty acids and the differentiation of adipocytes for energy storage (*Monsalve et al., 2013*; *Lamichane et al., 2018*; *Janani and Ranjitha Kumari, 2015*). Upon analysis of the DEGs (*Figure 3—source datas 1–3*), we found that most of the genes were related to fatty acid oxidation, which suggests that curcumin regulates gene transcription by activated PPARα in the liver.

We further confirmed the elevated mRNA expression levels of *Ppara*, *Apoa2*, *Apoa1*, and some genes related to fatty acid metabolism in curcumin-supplemented groups by real-time qPCR (*Figure 3e*). Although *Pparg* mRNA expression was confirmed to be upregulated by curcumin (*Figure 3—figure supplement 1*), its levels were much lower than *Ppara* in the liver (data not shown). These results suggest that curcumin is a PPARα/γ dual activator, and that the various physiological activities of curcumin may be derived from the complex regulation of PPARα and PPARγ. Because expression of PPARα and PPARγ varies largely in different tissues and organs, curcumin exhibits diverse physiological activities in different studies depending on the organs that are evaluated (*Monsalve et al., 2013*; *Lamichane et al., 2018*; *Harmon et al., 2011*), while *Ppara* should be the major target of curcumin owing to its abundance in mouse liver.

## PPARα levels in the liver were increased and showed more intranuclear localization in mice supplemented with curcumin

As a transcription factor, activated PPARα transferred to the nucleus and formed heterodimers with retinoid X receptor (RXR). The heterodimers bind to the peroxisome proliferator response element, a specific DNA sequence present in the promoter region of PPAR-regulated genes (*Xu et al., 2003*). To elucidate how curcumin affects PPARα, we evaluated intracellular localization and PPARα levels in the liver. Compared with mice without curcumin supplementation, the fluorescence signal of PPARα in the Cur and A-Cur groups was concentrated in the nucleus, resulting in a higher signal intensity (*Figure 4a*). Moreover, as shown in *Figure 4b*, PPARα protein levels were increased in curcumin-supplemented mice, which is consistent with real-time qPCR results. These results confirm that the synthesis and activation of PPARα are promoted by curcumin and also explain the DEGs related to lipid metabolism in the liver.

## Curcumin induced a higher abundance of peroxisomes and elevated a variety of peroxisome proteins downstream of PPARα

Proliferation of peroxisomes provides additional evidence that PPARα is a target protein of curcumin. The mitochondria and peroxisomes are the most important organelles responsible for fatty acid oxidation. However, very long chain fatty acids (VLCFAs) exhibiting >22 carbons are too long to be metabolized in the mitochondria, and must be metabolized in peroxisomes (*Islinger et al., 2018*). Activated PPARα is known to promote peroxisomes in mice and increase levels of fatty acid oxidation (*Islinger et al., 2018*; *Schrader et al., 2016*; *Lee et al., 1995*). We noticed that peroxisome is

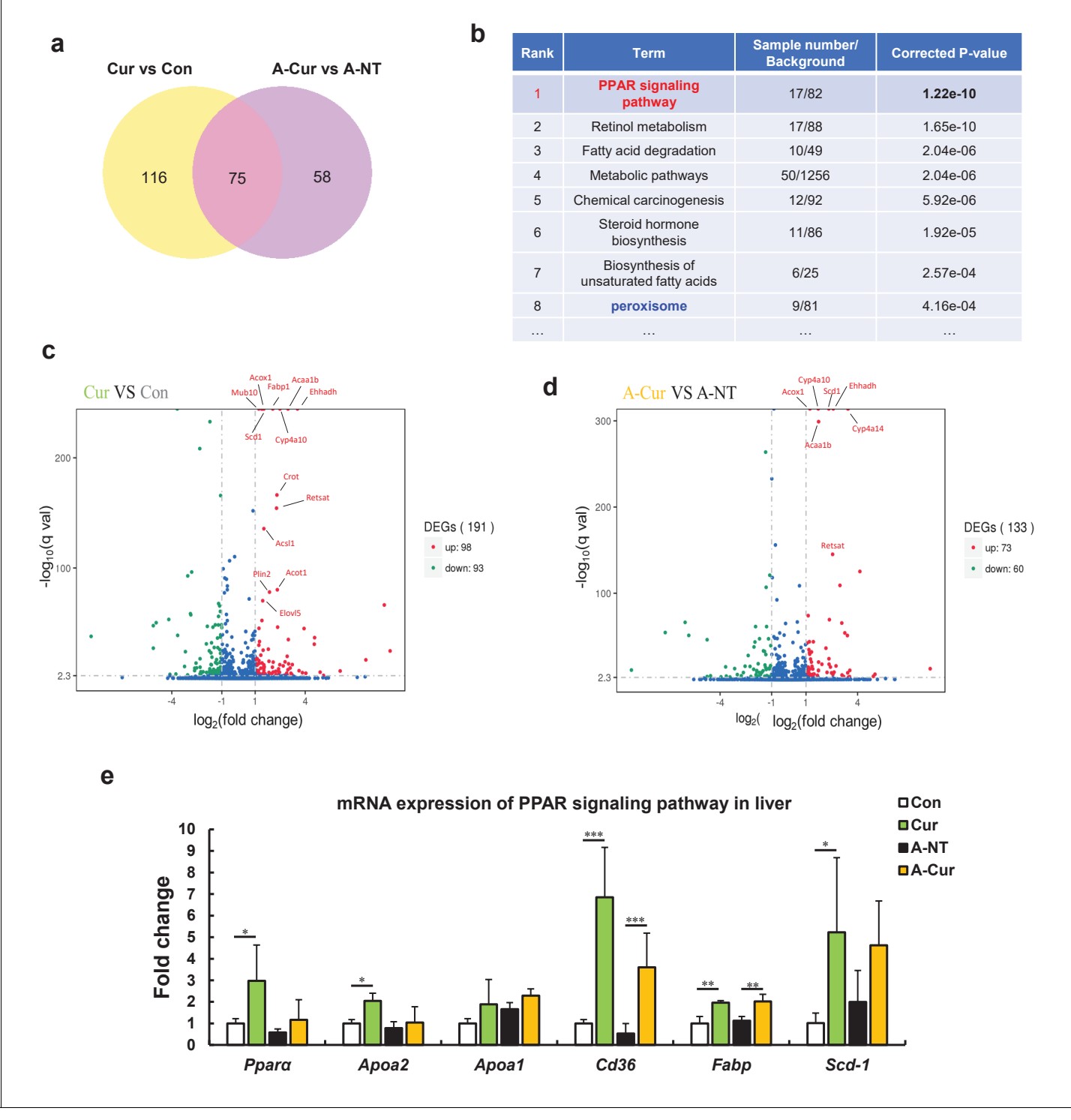

**Figure 3.** RNA sequence analysis showed that curcumin regulates many lipid metabolism-related genes via the peroxisome proliferator-activated receptor (PPAR) signaling pathway. (a) The Venn diagram shows that 75 genes are changed in the liver by supplementation with curcumin for 12 weeks. (b) Enrichment pathway analysis based on the KEGG database. (c and d) The Volcano plot diagram shows that differentially expressed genes (DEGs) affected by curcumin are related to lipid metabolism and the PPARa pathway. (e) Regulated genes were identified by real-time qPCR. Histograms show fold changes relative to the Con group. Data are mean ± SD. The Tukey–Kramer method was used for multiple comparisons of gene changes; *p < 0.05, **p < 0.01, ***p < 0.001.

The online version of this article includes the following source data and figure supplement(s) for figure 3:

**Source data 1.** Cur vs Con.DEG.

*Figure 3 continued on next page*

*Figure 3 continued*

**Source data 2.** ACur vs ANT.DEGs.
**Source data 3.** the list of 75 DEGs.
**Source data 4.** CurvsCon_all.DEG_KEGG_pathway_enrichment.
**Source data 5.** ACurvsANT_all.DEG_KEGG_pathway_enrichment.
**Figure supplement 1.** mRNA expression of Pparγ in the liver.

ranked 8 in the analysis of enrichment pathways (*Figure 3b*). Many enzymes related to fatty acid ß oxidation that are located in peroxisomes, including Acox, Acaa1, Ehhadh, Crat, and Crot, are known to be upregulated (*Figure 3—source datas 1* and *2*). Another important upregulated protein is Pex11, a protein that regulates peroxisome division to increase peroxisome abundance (*Weng et al., 2013*; *Knoblach and Rachubinski, 2010*).

Catalase is one of the most important enzymes involved in protecting the cell from oxidative damage by catalyzing the decomposition of hydrogen peroxide. Due to localization in peroxisomes, it is usually used as a marker for peroxisomes (*Goldman and Blobel, 1978*; *Fahimi, 1969*). We detected catalase in the liver by immunohistochemistry (IHC) and western immunoblotting to confirm peroxisome abundance. Results obtained for catalase by IHC and immunoblotting suggest a

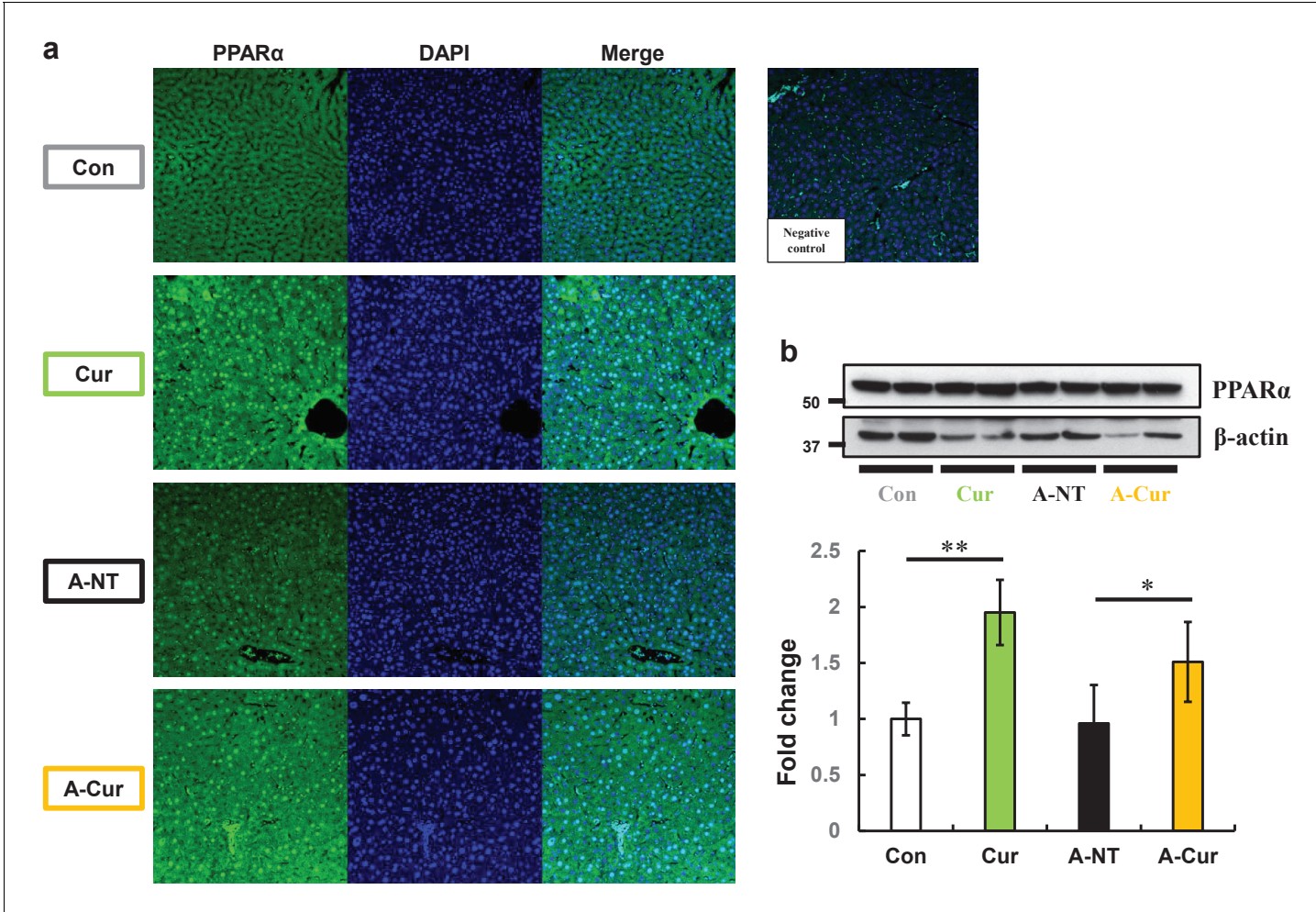

**Figure 4.** PPARα levels in the liver are increased and show more intranuclear localization in curcumin-supplemented mice. (**a**) Immunofluorescence results of PPARa showed obvious intranuclear signal accumulation and extranuclear signal enhancement after curcumin ingestion. (**b**) Levels of PPARa proteins in the liver were determined by western immunoblot. Data are mean ± SD. The Tukey–Kramer method was used for multiple comparisons; *p < 0.05, **p < 0.01.

higher abundance of peroxisomes after curcumin supplementation (*Figure 5a and b*). The overexpression of catalase also reflects an increase in fatty acid metabolic activity and oxidative stress in hepatocytes.

In addition, we unexpectedly found that the protein band observed at a molecular weight of 75 kD is significantly increased in SDS PAGE of the liver extracts stained by Coomassie brilliant blue (*Figure 5c*). We analyzed the proteins in this band by proteomic LC-MS/MS analysis and determined that it was comprised of several peroxisomal proteins related to fatty acid oxidation, including Ehhadh, Hsd17b4, and Acsl1 (*Figure 5d*, *Supplementary file 2*). These results demonstrate that curcumin regulates peroxisome abundance via PPARα activation.

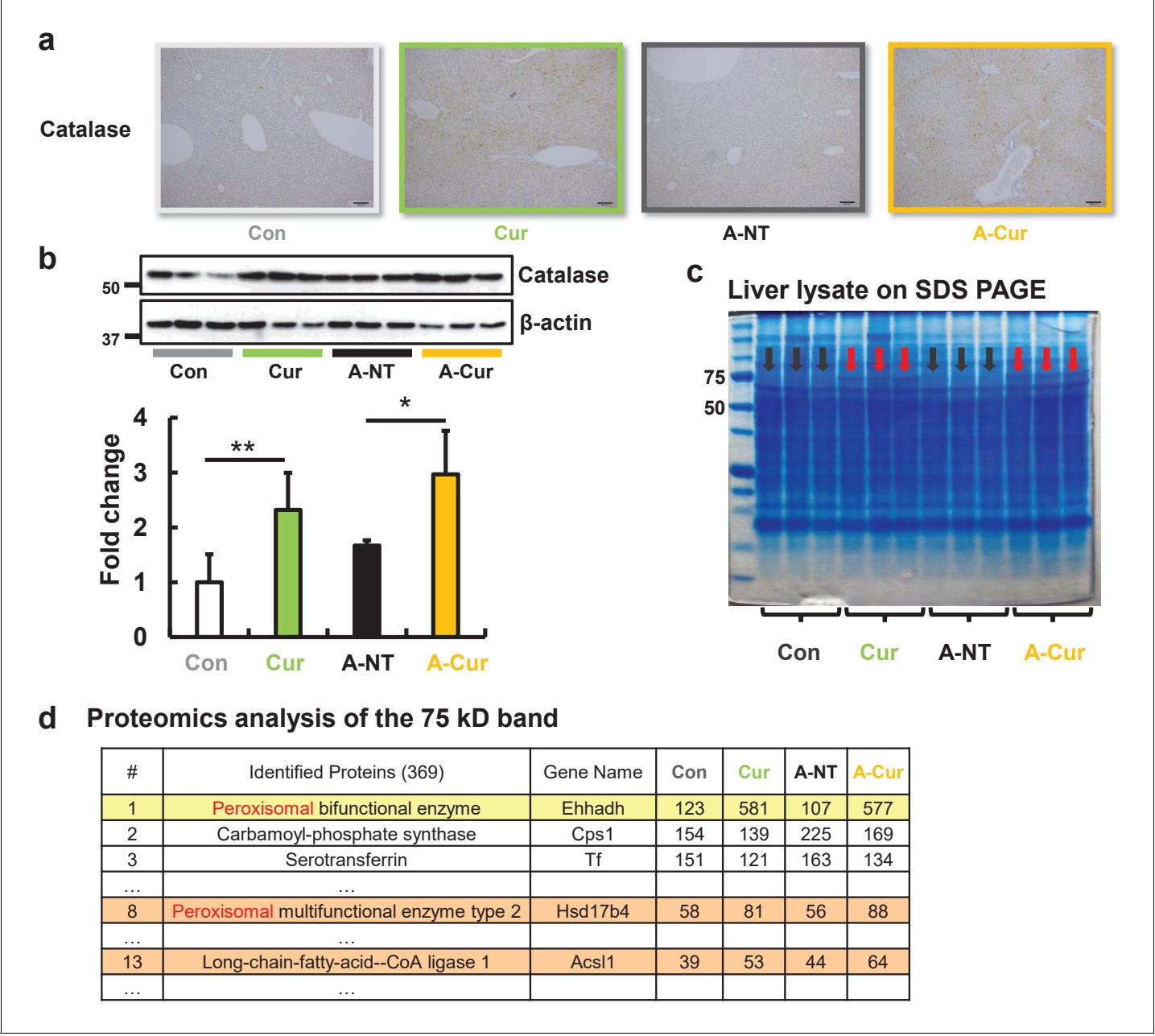

**d  Proteomics analysis of the 75 kD band**

| # | Identified Proteins (369) | Gene Name | Con | Cur | A-NT | A-Cur |
|---|---|---|---|---|---|---|
| 1 | Peroxisomal bifunctional enzyme | Ehhadh | 123 | 581 | 107 | 577 |
| 2 | Carbamoyl-phosphate synthase | Cps1 | 154 | 139 | 225 | 169 |
| 3 | Serotransferrin | Tf | 151 | 121 | 163 | 134 |
| ... | ... | | | | | |
| 8 | Peroxisomal multifunctional enzyme type 2 | Hsd17b4 | 58 | 81 | 56 | 88 |
| ... | ... | | | | | |
| 13 | Long-chain-fatty-acid--CoA ligase 1 | Acsl1 | 39 | 53 | 44 | 64 |
| ... | ... | | | | | |

**Figure 5.** Peroxisome and fatty acid oxidative enzymes are increased in the liver. IHC (**a**) and western blot (**b**) results confirmed that catalase was significantly increased after curcumin supplementation. (**c**) The protein band near 75 kD of liver lysates exhibited different expression levels for the commercial diet and curcumin diet groups. (**d**) Three peroxisomal proteins were identified in the band that were found to be upregulated by LC-MS/MS. Data are mean ± SD. The Tukey–Kramer method was used for multiple comparisons; *p < 0.05, **p < 0.01.

## Discussion

In previous studies, curcumin was found to exert physiological activities involved in the regulation of several transcription factors (PPARr, NFκB, AP-1, STAT, etc.) and their signaling pathways (*Liczbiński et al., 2020*; *Shishodia, 2013*). In the present experiment, we found that the gene expression changes in mouse liver after curcumin supplementation are centered on the PPAR signaling pathway. In the enrichment pathway analysis, most of the DEGs involved in retinol metabolism, metabolic pathways, fatty acid degradation, and the peroxisome pathway are also downstream of PPARα. Until now, research on curcumin has mainly focused on the activation of PPARr, and it has been demonstrated that curcumin participates in glucose and lipid metabolism, inflammatory cytokine production, and inhibiting tumor cell proliferation via PPARr (*Shishodia, 2013*). Although some studies have found that PPARα expression is increased after curcumin intake (*Kong et al., 2020*; *Zeng et al., 2016*), there is currently a lack of further experimental results revealing any connection between curcumin and PPARα pathway activation. Our results complement the theoretical system suggesting that curcumin regulates the transcription of many genes in the liver depending centrally on PPARα activation and broadly affects fatty acid transport and catabolism. These changes in genes and proteins levels may be involved, in which curcumin regulates the occurrence and development of various diseases or phenotypes, such as amyloidosis, changes in HDL and triglycerides, oxidative stress, peroxisome proliferation, and hepatocyte hypertrophy (*Figure 6*).

Due to the unexpected promotion of AApoAII amyloid deposition by curcumin in this study, we suspect that the anti-amyloid effect of curcumin is not applicable to all types of amyloidosis. To better understand the findings of the present study, we suggest that the effects of curcumin on progression of amyloidosis should be divided into two aspects, one of which is a pro-amyloidosis effect. In this experiment, the plasma concentration of ApoA-II increased almost threefold (*Figure 2a*) via PPARα pathway activation. Levels of amyloid protein are the most important factor for progression of amyloidosis, and suppression of amyloid protein levels is the main target for treatment of various systemic amyloidosis, including AA, AL, ATTR, and dialysis-related Aβ$_2$M amyloidosis (*Merlini and Bellotti, 2003*; *Knowles et al., 2014*; *Calhoun et al., 1999*; *Zhang et al., 2010*; *Ueda et al., 2007*).

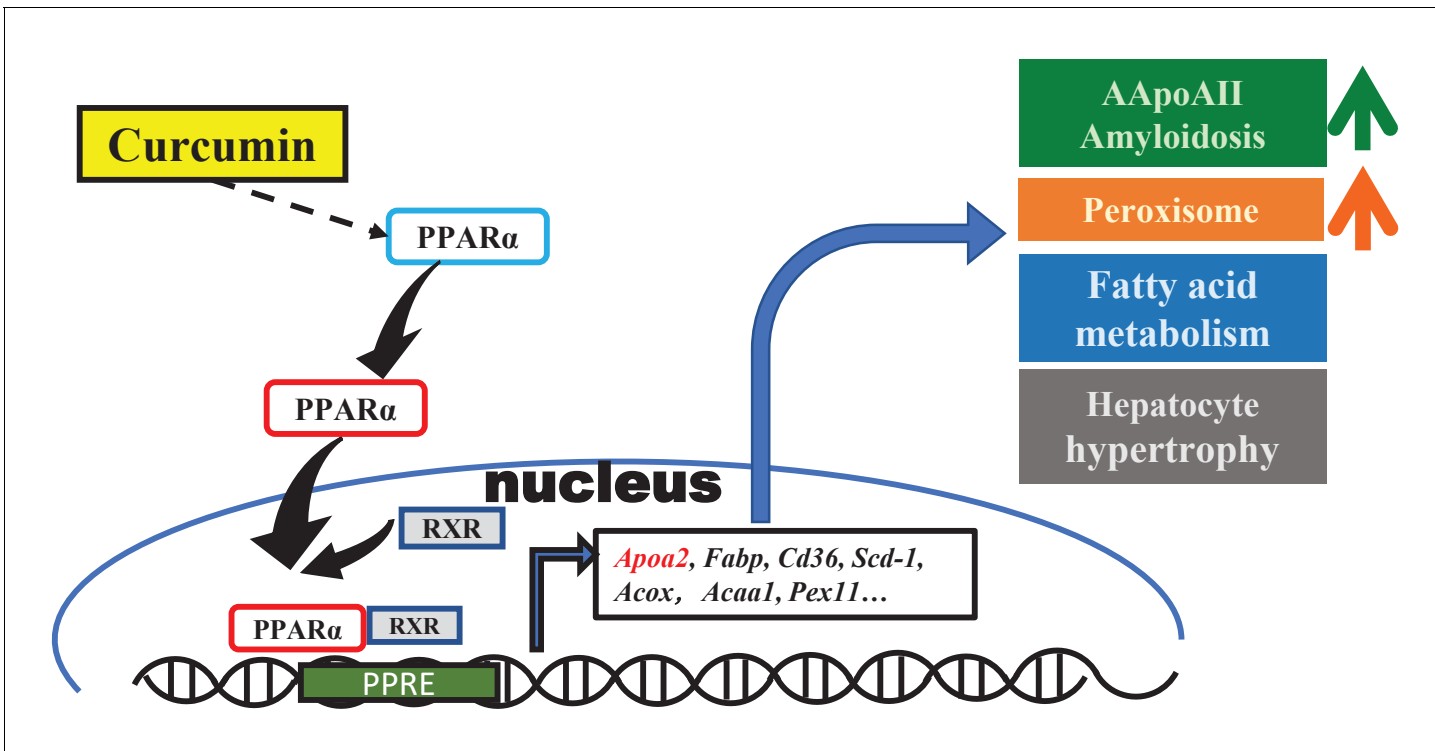

**Figure 6.** Schematic diagram of the effect of curcumin mediated by PPARa in mouse liver. Curcumin regulates gene expression via PPARa activation and exhibits biological activities involved in amyloidosis, peroxisome proliferation, lipid metabolism, and hepatocyte hypertrophy.

In ApoA-II transgenic mice, the serum concentration of ApoA-II increased twofold and AApoAII amyloid deposition was notably accelerated (*Ge et al., 2007*). However, calorie restriction (diet) decreased the ApoA-II/ApoA-I ratio in serum and suppressed amyloidosis (*Li et al., 2017*). Thus, we believe that the pro-amyloid effect of curcumin observed here is mainly caused by increased ApoA-II levels.

The other aspect is the anti-amyloid effect of curcumin. Curcumin has shown general affinity for the amyloid protein structure and it has been demonstrated that this binding is efficient to maintain stability of the amyloid protein and inhibit the formation of insoluble amyloid fibrils (*Stefani and Rigacci, 2013*; *Ahmad et al., 2017*; *Hafner-Bratkovic et al., 2008*). In addition to direct binding to amyloid proteins, it has also been shown that curcumin activates autophagy and macrophages in some studies to reduce the dysfunction caused by amyloid proteins (*Zhang et al., 2006*; *Li et al., 2014*). Moreover, it was reported that some molecules elevate lysosome biosynthesis via activation of PPARα and accelerate the clearance of amyloid proteins and protein aggregates (*Ghosh et al., 2015*; *Chandra et al., 2018*). Compared with previous data, we noticed that amyloid deposition in *Apoa2^c* transgenic mice with twice the serum concentration of ApoA-II is more severe than in those with greater ApoA-II concentrations in this study (*Ge et al., 2007*). In transgenic mice injected with 1 µg AApoAII fibrils for 8 and 12 weeks, the average amyloid score in the liver and spleen at 12 weeks and the amyloid index at 8 weeks was 3.4, 4, and 3, respectively, compared with values of 2, 3.4, and 1.5, respectively, in curcumin-supplemented mice. We believe that this difference demonstrates that curcumin may also exert a certain anti-amyloid ability in this experiment, but it is difficult to evaluate and further analyses should be carried out in the future.

Curcumin interacts with various cellular metabolic pathways by activating PPARs. Most biological process, such as metabolic control and defect, involve complex molecular interactions and are regulated via various signaling pathways. When amyloid proteins play functional roles in certain metabolic pathways, progression of amyloidosis may be accelerated or decelerated via treatments that modulate such metabolic pathways. As ApoA-II interacts particularly with lipid metabolism, it is likely that amyloid deposition was increased by metabolic changes caused by curcumin, including acceleration of ß-oxidation of fatty acids, reduction of lipogenesis, and increased synthesis of apolipoproteins.

In addition, we think the peroxisome proliferation is the most important factor that explains the hepatocyte and liver hypertrophy. There was no significant change in food intake between the four groups investigated in our experiments (*Figure 1—figure supplement 2*), and no significant lipid deposition was observed in the liver (*Figure 1—figure supplement 4*). Peroxisomes in liver parenchymal cells are very few and represent <2% of cytoplasmic volume under physiological conditions and contribute about 35% of the $H_2O_2$-production levels. However, peroxisomes in the liver in the presence of peroxisome agonists may occupy as much as 25% of the cytoplasmic volume (*Yeldandi et al., 2000*). In previous studies, long-term administration of PPARα agonists was shown to induce disordered peroxisome proliferation, liver hypertrophy, and liver tumors in mice (*Corton et al., 2018*). In our experiments, overexpression of catalase and peroxisomal proteins suggests that curcumin promotes peroxisome proliferation mediated by PPARα. The hypertrophic changes in the hepatocyte nucleus we observed in histological sections are consistent with the early pathological changes of hepatocyte heterogeneity.

Peroxisomes are a conserved organelle and play a key role in lipid metabolism and redox homeostasis in both plants and mammals (*Yeldandi et al., 2000*; *Farré et al., 2019*; *Schrader et al., 2020*). According to the proliferation mode of peroxisomes, the process of peroxisome fission is a three-step process involving peroxisome elongation, constriction, and scission. Pex11 is essential for this process and its overexpression causes peroxisome proliferation, while its deletion causes a decrease in the number of peroxisomes (*Erdmann and Blobel, 1995*). Overexpression of Pex11 in curcumin-supplemented mice suggests that Pex11 may play a key role in peroxisome proliferation mediated by PPARα and curcumin. Moreover, proteins such as Acsl1, Acaa1, and Ehhadh upregulated by curcumin in this experiment are localized in the peroxisome matrix and may be involved in this process.

A hallmark of eukaryotic cells is the presence of membrane-bound organelles, such as endoplasmic reticulum (ER), mitochondria, and peroxisomes. Such distinct compartments create special micro-environments for more efficient metabolic reactions. To coordinate complex metabolic processes and signal transduction, there are functional interplays between various organelles. Due to

the central metabolic role, it was shown that peroxisomes interact with many organelles involved in cellular lipid metabolism, such as the ER, mitochondria, lysosomes, and lipid droplets (*Schrader et al., 2015*; *Wanders et al., 2018*; *Sugiura et al., 2017*). There is also functional inter-play between peroxisomes and the nucleus, which may also involve signaling via $H_2O_2$ (*Schrader et al., 2013*; *Mullineaux et al., 2018*).

Other means by which peroxisome proliferation affects ER or mitochondrial functions are also known. However, the mechanism by which curcumin activates PPARs is not yet clear. There are two means of activating PPARs, namely ligand-dependent and ligand-independent. In the ligand-dependent manner, the molecular shape of PPARs is modified by ligand binding in the cytoplasm, and PPARs enter into the nucleus. In a ligand-independent manner, PPARs can be phosphorylated by protein kinases to induce a structural change of phosphorylated PPARs, even in the absence of ligands (*Lamichane et al., 2018*; *Harmon et al., 2011*; *Magadum and Engel, 2018*). Further experiments are needed to confirm whether curcumin directly binds and activates PPARs as an exogenous ligand, or whether activation involves a ligand-independent pathway.

PPARα and PPARr involve different aspects of metabolic pathways, such as decomposition or storage of fatty acids, fatty acid-based energy production, or glucose-based energy production. PPARα agonists (e.g., fibrates) or PPARr agonists (e.g., thiazolidinedione) play important roles in the treatment of hyperlipidemia and type 2 diabetes in the clinic. PPARα/r dual agonists are also under development to treat more complex metabolic diseases, but some exhibit side effects and cause liver or cardiac dysfunction (*Kim et al., 2019*; *Kalliora et al., 2019*). In clinical trials, it has been demonstrated that supplementation with curcumin at a high dose is safe in humans (*Sahebkar, 2014*; *Chen et al., 2015*). Improving the molecular structure of drugs based on that of curcumin offers the possibility to produce dual or specific agonists without side effects.

Taken together, our results demonstrate the novel agonistic effect of curcumin on PPARα. We identified specific effects of curcumin on mice, including promotion of AApoAII amyloidosis and peroxisome proliferation. Curcumin is involved in various physiological activities mediated by PPARs activation, leading to regulation of genes participating in the PPAR pathway. The beneficial use of curcumin based on these particular abilities requires further consideration. The development of derivative agents based on curcumin with high bioavailability or specific effects may have far-reaching significance for the treatment of diseases such as amyloidosis, hyperlipidemia, type 2 diabetes, and other metabolic disorders.

## Materials and methods

### Animals and drug administration

R1.P1-*Apoa2^c* congenic mice were used in this study, which carry the amyloidogenic type c allele (*Apoa2^c*) of amyloidosis-susceptible SAMP1 strain on a genetic background of the SAMR1 strain. R1.P1-*Apoa2^c* mice exhibit a normal aging process and develop accelerated AApoAII amyloidosis by oral or intravenous administration of AApoAII fibrils (*Higuchi et al., 1995*). Mice were maintained under SPF conditions at 24 ± 2°C with a light-controlled regimen (12 hr light/dark cycle) in the Division of Animal Research, Research Center for Supports to Advanced Science, Shinshu University. The mice were fed a commercial diet (Con group and A-NT group) or curcumin diet (Cur group and A-Cur group) and tap water ad libitum. The commercial diet is a MF diet (Oriental Yeast, Tokyo, Japan) and the curcumin diet is the MF diet supplemented with 0.5% or 2% w/w curcumin (Wako, Osaka, Japan).

Three to five R1.P1-*Apoa2^c* congenic mice were housed in a single cage. Female mice were used for experiments to avoid the anticipated adverse impacts due to fighting among male mice. Mice were sacrificed by cardiac puncture under deep sevoflurane anesthesia after 8 weeks and 12 weeks of curcumin intake. Plasma and half of the major organs (heart, liver, spleen, stomach, small intestine, tongue, skin, lung, and kidney) were snap-frozen by liquid nitrogen and stored at −80°C for biochemical analysis. The remaining organs were fixed in 10% neutral buffered formalin followed by embedding in paraffin for histochemical analysis. All experiments were approved by the Committee for Animal Experiments of Shinshu University (Approval No. 280086).

## Induction of AApoAII amyloidosis

AApoAII amyloid fibrils were isolated using Pras' method (*Pras et al., 1969*) from the livers of R1.P1-*Apoa2^c* mice having severe amyloid deposits. Mice in the amyloid-induced groups were injected with 1 µg amyloid fibrils into the tail vein for induction of AApoAII amyloidosis at 8 weeks of age. AApoAII fibrils were sonicated before use and the injection was performed immediately.

## Evaluation of amyloid deposition

Amyloid deposits were detected in paraffin organ sections stained with a saturated solution of 1% Congo red dye. An amyloid score (from 0 to 4) in each organ was determined semi-quantitatively as described previously (*Xing et al., 2002*) under polarizing light microscopy (LM) (Axioskop 2, Carl Zeiss, Tokyo, Japan). Two observers, with no information of the Congo red stained tissues, graded the degree of amyloid deposition in each mouse, separately. The degree of amyloid deposition in each mouse was represented by an amyloid index (AI), which is the average of the amyloid scores in seven organs (heart, liver, spleen, stomach, small intestine, tongue, and skin).

## Hepatocyte size in each group

To analyze the hepatocyte size in each mouse quantitatively, we captured five images of each section selected randomly at 400× magnification and determined the average hepatocyte size in each image by calculating the total area divided by the cell counts using an image processing program (NIH ImageJ software, version 1.61). We determined the average hepatocyte size of each mouse and then performed a statistical analysis between all four groups.

## Immunohistochemistry and immunofluorescence analysis

We detected AApoAII deposition and catalase by immunohistochemistry (IHC) following a previously described method (*Li et al., 2017*). Antiserum against mouse ApoA-II was produced against guanidine hydrochloride-denatured AApoAII in our laboratory (*Higuchi et al., 1983*) and applied at a dilution ratio of 1:3000. Catalase antibody was applied (1:500, GTX110704, GeneTex Inc, CA, USA) to reveal the degree of peroxisome change in the liver. After incubation overnight at 4°C with the primary antibody, the sections were incubated with the biotinylated secondary antibody (1:300, DAKO, Glostrup, Denmark) for 1 hr at room temperature. Target proteins were identified by the horseradish peroxidase-labeled streptavidin-biotin method (1:300, DAKO). In the immunofluorescence experiments, the sections were incubated with the PPARα antibody (1:500, GTX101098, GeneTex Inc, CA, USA) overnight and incubated with Alexa Fluor 488 goat anti-rabbit antibody (1:500, Thermo Fisher Scientific, Japan) for 1 hr at room temperature and incubated with DAPI for 10 min. Images were captured immediately using a confocal laser fluorescence microscope (LSM 880 with Airyscan, Carl Zeiss, Germany). In a negative control section, the primary antibody was omitted to confirm the specificity of staining.

## Lipid metabolism analysis

Lipid metabolism levels were determined using quantitative assay kits by means of HDL cholesterol, total cholesterol, and triglyceride concentrations in the plasma with the instructions provided by the manufacturer (HDL-cholesterol E test, 431–52501; Total-cholesterol E test, 439–17501; TG E test, 432–40201, FUJI FILM Wako, Osaka, Japan). Each sample repeated at least three times.

## AST and ALT detection

Two hundred microliters of frozen mouse plasma per mouse was sent to Nagahama life science laboratory (Oriental Yeast, Tokyo, Japan) for determination of AST and ALT levels. The laboratory provided a test report.

## Immunoblot analysis

We measured proteins levels by western blotting as described previously (*Ge et al., 2007*; *Li et al., 2017*). To determine plasma levels of ApoA-II, ApoA-I, and ApoE, 0.5 µL samples from each mouse were separated by Tris-Tricine/SDS–16.5% or 15% polyacrylamide gel electrophoresis (PAGE). After electrophoresis, proteins were transferred to a polyvinylidene difluoride (PVDF) membrane (Immobilon, 0.2 µm pore, Millipore Corp., MA, USA) and incubated overnight at 4°C with primary antibody

solution containing polyclonal rabbit anti-mouse ApoA-II antiserum (diluted 1:3000) or the ApoA-I antiserum (diluted 1:4000) produced in our laboratory, or ApoE antibody (1:500, Santa Cruz, San Francisco, CA, USA). Next, horseradish peroxidase-conjugated anti-rabbit IgG (Code #7074, Cell Signaling Technology Inc, Danvers, MA, USA) (1:3000) was used for 1 hr incubation at room temperature and target proteins were detected by the enhanced chemiluminescence (ECL) method. Thirty micrograms of liver lysates were separated on Tris-Tricine/SDS–12% PAGE to determine levels of PPARa (1:3000, GTX101098, GeneTex Inc), β-actin (1:3000, GTX110564, GeneTex Inc), and catalase (1:3000, GTX110704, GeneTex Inc, CA, USA). Target protein levels were analyzed using the NIH ImageJ software.

## RNA sequence analysis

We selected liver samples from the 12-week group for RNA sequence analysis, which exhibit more obvious amyloid deposition in amyloid-induced animals and no abnormal changes in AST and ALT levels. Ten milligrams of each mouse liver stored at −80℃ was homogenized in TRIzol RNA isolation reagent (Invitrogen-Thermo Fisher, Tokyo Japan) and pooled into four sample tubes (Con, Cur, A-NT and A-Cur groups, N = 3–5), and the samples were sent to Filgen (Nagoya, Japan). Total RNA was extracted and the RNA purity and integrity were confirmed using a Bioanalyzer 2100 system (Agilent Technology, Santa Clara, CA USA). mRNA sequencing analysis was performed with an Illumina next generation sequencing platform. Sequencing count data were analyzed using the DESeq2 software to determine the significant DEGs among the different groups. Biological functions into which DEGs accumulated were analyzed using an annotation database (Gene Ontology and KEGG Pathway Database) to elucidate the mechanism of the effects of curcumin supplementation.

## Gene expression analysis

We followed a previously described method to confirm mRNA expression levels (*Tian et al., 2014*). Quantitative real-time qPCR analysis was carried out using an ABI PRISM 7500 Sequence Detection system (Applied Biosystems, New York, USA) with SYBR Green (TaKaRa Bio, Tokyo, Japan). The β-actin gene was used to normalize gene expression. Each sample repeated at least three times. The forward and reverse primer sequences for real-time PCR are listed in *Supplementary file 3*. Chemical reagents used in the experiments, unless otherwise specified, were obtained from Wako Pure Chemical Industries Ltd. (Osaka, Japan).

## Nano-flow liquid chromatography-ion trap mass spectrometry (LC-MS/MS)

Thirty micrograms of liver lysates were separated on Tris-Tricine/SDS–12% PAGE and stained with Coomassie brilliant blue for 20 min. The stained bands near 75 kD were excised and soaked in 50 mM Tris-HCl, pH 8.0, containing 50% acetonitrile for 30 min. The gel was dried in a Speed-Vac (Savant) and incubated in 50 mM triethylammonium bicarbonate containing proteomics grade trypsin (Sigma-Aldrich, Tokyo, Japan) at 37℃ for 20 hr. The digests were extracted from the gel with 100–200 μL of 0.1% TFA containing 60% acetonitrile. These extracts were evaporated in a Speed-Vac and stored at −80℃ until assayed.

Samples were resuspended in 0.1% formic acid and introduced into a nano-flow HPLC system, EASY-nLC 1200 (Thermo Fisher Scientific Inc, Waltham, MA, USA). A packed nano-capillary column, NTCC-360/75-3-123 (0.075 mm I.D. × 125 mm L, particle diameter 3 μm, Nikkyo Technos Co., Ltd., Tokyo, Japan), was used at a flow rate of 300 nL/min with a 2–80% linear gradient of acetonitrile for 80 min. Eluted peptides were directly detected with an ion trap mass spectrometer (QExactive HF; Thermo Fisher Scientific Inc, Waltham, MA, USA). For ionization, a spray voltage of 2.0 kV and capillary temperature of 250℃ were used. The mass acquisition method consisted of one full MS survey scan with an Orbitrap resolution of 60,000, followed by an MS/MS scan of the most abundant precursor ions from the survey scan with an Orbitrap resolution of 15,000. Dynamic exclusion for the MS/MS was set to 30 s. An MS scan range of 350–1800 m/z was employed in the positive ion mode, followed by data-dependent MS/MS using the HCD operating mode on the top 15 ions in order of abundance. The data were analyzed with Proteome Discoverer (Thermo Fisher Scientific Inc, Waltham, MA, USA), Mascot software (Matrix Science Inc, Boston, MA, USA), and Scaffold software (Proteome Software, Inc, Oregon, USA). Swissprot and GenBank databases were used.

## Statistical analyses

For comparison of parametrical data, one-way analysis of variance (ANOVA) with Tukey's test was performed using the SPSS 26.0 software package (Abacus Concepts, Berkley, CA USA). For comparison of nonparametric data, the Kruskal–Wallis test with the Steel–Dwass test was performed using the R software version 3.4.3. p-values <0.05 were considered to be statistically significant.

## Acknowledgements

This work was supported in part by Grants-in-Aid for Scientific Research (B) 17H04063 and Challenging Exploratory Research 26670152 from the Ministry of Education, Culture, Sports, Science and Technology, Japan, The authors thank Drs. Kiyoshi Matsumoto and Takahiro Yoshizawa, Hiroto Yamanaka, and Ms. Kayo Suzuki (Research Center for Supports to Advanced Science, Shinshu University) for animal care and technical assistance in the preparation of tissue sections.

## Additional information

### Funding

| Funder | Grant reference number | Author |
|---|---|---|
| Ministry of Education, Culture, Sports, Science and Technology | 17H04063 | Jian Dai |
| Ministry of Education, Culture, Sports, Science and Technology | 26670152 | Jian Dai |

The funders had no role in study design, data collection and interpretation, or the decision to submit the work for publication.

### Author contributions

Jian Dai, Conceptualization, Resources, Data curation, Software, Formal analysis, Investigation, Methodology, Writing - original draft, Project administration; Ying Li, Fuyuki Kametani, Xiaoran Cui, Yuichi Igarashi, Jia Huo, Data curation, Formal analysis, Investigation, Writing - review and editing; Hiroki Miyahara, Formal analysis, Writing - review and editing; Masayuki Mori, Formal analysis, Methodology, Writing - review and editing; Keiichi Higuchi, Conceptualization, Writing - original draft, Writing - review and editing

### Author ORCIDs

Jian Dai (iD) https://orcid.org/0000-0001-8097-6756

### Ethics

Animal experimentation: All experiments were approved by the Committee for Animal Experiments of Shinshu University (Approval No. 280086). Mice were sacrificed by cardiac puncture under deep sevoflurane anesthesia, and every effort was made to minimize suffering.

### Decision letter and Author response

Decision letter https://doi.org/10.7554/eLife.63538.sa1
Author response https://doi.org/10.7554/eLife.63538.sa2

## Additional files

### Supplementary files

- Supplementary file 1. Liver and body weight.
- Supplementary file 2. 70KD-proteomics.
- Supplementary file 3. Primers.

• Transparent reporting form

## Data availability

The data used to support the findings of this study are included within the article. Key RNA seq analysis data is available on Dryad (https://doi.org/10.5061/dryad.9ghx3ffgm). All other data supporting the findings of this study will be made available upon reasonable request to the corresponding authors.

The following dataset was generated:

| Author(s) | Year | Dataset title | Dataset URL | Database and Identifier |
|---|---|---|---|---|
| Dai J, Li Y, Kametani F, Cui X, Igarashi Y, Miyahara H, Mori M, Higuchi K | 2021 | The results of RNA seq analysis (Curcumin promotes progression of AApoAII amyloidosis and peroxisome proliferation in mice by activating the PPARα signaling pathway) | https://doi.org/10.5061/dryad.9ghx3ffgm | Dryad Digital Repository, 10.5061/dryad.9ghx3ffgm |

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
