## [Decision Letter]

**Acceptance summary:**

We believe this study sheds a new light on how curcumin can have opposite effects on different types of amyloidosis. These are very important findings, especially for those affected with AApoAII amyloidosis, who might otherwise take curcumin based on it's property of reducing other types of amyloid. The additional data of the mechanism of action on PPAR-α is also very interesting.

**Decision letter after peer review:**

Congratulations, we are pleased to inform you that your article, "Curcumin promotes AApoAII amyloidosis and peroxisome proliferation in mice by activating the PPARα signaling pathway", has been accepted for publication in *eLife*.

Reviewer #1:

Here, a mouse model of amyloid apolipoprotein A-II (AApoAII) to test effects of a curcumin-supplemented diet (12 weeks) on AApoAII amyloid deposition The authors report a significantly increase in AApoAII amyloid deposition especially in the liver and spleen. RNA-sequence analysis revealed that the molecular pathway involves PPARα activation leading to increased Apoa2 mRNA expression. These are very important findings, especially for those affected with this disorder, who might otherwise take curcumin based on it's know property of reducing amyloid formation. These important results suggest quite the opposite.

Figure 2: The authors need to show more of the Westerns in part A. They are cropped much too tightly to get a true sense of size and intensity of the bands. Also please show controls for load for each blot.

Figure 4: Same issue as Figure 2. The authors need to show more of the Westerns in part B. They are cropped much too tightly to get a true sense of size and intensity of the bands.

Reviewer #2:

Many previous papers have demonstrated that curcumin inhibits amyloidogenesis in vitro and in mouse models. The authors found the opposite in the case of seeded ApoAII amyloidosis. This is a strong *eLife* paper because they figured out why, which has implications way beyond amyloid diseases. The authors establish that curcumin functions predominantly by activating PPAR-α signaling in the liver. 75 genes are upregulated in mouse liver upon Curcumin treatment, including ApoAII. That Curcumin promotes ApoAII amyloidosis and peroxisome proliferation in mice by activating PPAR-α signaling pathway is a secondary strong reason that this paper merits publication.

This paper requires re-evaluation of the mechanism by which Curcumin inhibits amyloid formation in previously published mouse models, unlike direct amyloid binding invoked previously, it most likely functions via a transcriptional program.

Scientists who now want to activate PPAR-α signaling in vivo have a small molecule to do so.

Well-written!

While the authors are following the Amyloid nomenclature committee rules, please drop the extra A in the title so people don't get confused that the paper is about AA amyloidosis.

Also clarify via transcriptional program how selective the PPAR-α vs the PPAR other signaling that Curcumin is activating at the utilized dose.

Reviewer #3:

The authors have clearly demonstrated the mechanism by which a molecule considered a generic anti-amyloid can have an opposite effect on a specific type of amyloidosis.

This discovery has a great relevance because will stimulate the research for discovering anti amyloid drugs highly specific for every type of amyloidosis

Wonderful work!

I'm wondering if the concentration of APOC II and III were determined. (It's possible that this is my fault and I missed the data) If not, it would be nice to have these data and see how the induction on peroxisome affect the expression of these two lipoproteins.

I have the curiosity to know if the proteome of the amyloid deposits changed under the effect of curcumin.

---

## [Author Response]

Reviewer #1:Here, a mouse model of amyloid apolipoprotein A-II (AApoAII) to test effects of a curcumin-supplemented diet (12 weeks) on AApoAII amyloid deposition The authors report a significantly increase in AApoAII amyloid deposition especially in the liver and spleen. RNA-sequence analysis revealed that the molecular pathway involves PPARα activation leading to increased Apoa2 mRNA expression. These are very important findings, especially for those affected with this disorder, who might otherwise take curcumin based on it's know property of reducing amyloid formation. These important results suggest quite the opposite.Figure 2: The authors need to show more of the Westerns in part A. They are cropped much too tightly to get a true sense of size and intensity of the bands. Also please show controls for load for each blot.Figure 4: Same issue as Figure 2. The authors need to show more of the Westerns in part B. They are cropped much too tightly to get a true sense of size and intensity of the bands.

Thank you very much for your suggestion. We edited the Western blot images and supplemented the CBB stain of plasma protein to show the load (Figure 2—figure supplement 1D).

Reviewer #2:This paper requires re-evaluation of the mechanism by which Curcumin inhibits amyloid formation in previously published mouse models, unlike direct amyloid binding invoked previously, it most likely functions via a transcriptional program.Scientists who now want to activate PPAR-α signaling in vivo have a small molecule to do so.While the authors are following the Amyloid nomenclature committee rules, please drop the extra A in the title so people don't get confused that the paper is about AA amyloidosis.Also clarify via transcriptional program how selective the PPAR-α vs the PPAR other signaling that Curcumin is activating at the utilized dose.

1) Regarding the nomenclature used for AApoAII amyloidosis, we followed the recent version of the nomenclature decided on by the International Society of Amyloidosis (ISA) at the 2018 International Amyloid Symposium. Reference 1: Amyloid nomenclature 2018: recommendations by the International Society of Amyloidosis (ISA) nomenclature committee.

2) In our study, PPARα was selected as the major target protein of curcumin supplementation in the liver, rather than other PPARs, for the following reasons:

i) In previous studies, it has been reported that intake of PPARα agonists increases the plasma concentration of HDL and ApoA-II. In our study, we found a 2-fold increase in plasma levels of ApoA-II, and RNA seq results confirmed activation of the PPAR signaling pathway to be the most important molecular event following curcumin supplementation.

ii) PPARα, PPARβ, and PPARγ are functionally different. PPARα is a core transcription factor involved in lipid catabolism, while PPARγ plays an important role in the synthesis and storage of lipids in adipose tissue. We analyzed differentially expressed genes (DEGs) of the RNA seq analysis and found that a large number of DEGs located downstream of PPARα were up-regulated.

iii) It is known that activation of PPARα, but not PPARβ and PPARγ, leads to peroxisome proliferation and hepatocyte hypertrophy, and even hepatic cancer in rodents. The same phenomenon also appeared in our experiments.

The mechanism by which curcumin activates PPARs remains unclear. Further experiments are needed to confirm whether curcumin directly binds and activates PPARs as an exogenous ligand, or whether activation involves a ligand-independent pathway.

Reviewer #3:I'm wondering if the concentration of APOC II and III were determined. (It's possible that this is my fault and I missed the data) If not, it would be nice to have these data and see how the induction on peroxisome affect the expression of these two lipoproteins.I have the curiosity to know if the proteome of the amyloid deposits changed under the effect of curcumin.

Thank you for your constructive suggestions. In our study, we did not identify changes in the expression of ApoC-II and ApoC-III in the plasma. We confirmed the original RNA seq analysis data, and found no significant difference in the mRNA expression of *Apoc2* and *Apoc3* in the liver. In a previous study, we determined the proteomics of AApoAII deposition and did not detect ApoC-II and ApoC-III in the liver extract. Reference: Comprehensive proteomic profiles of mouse AApoAII amyloid fibrils provide insights into the involvement of lipoproteins in the pathology of amyloidosis.

As a result of the short amyloid induction time in the present study, the degree of amyloid deposition in the liver was still relatively low, especially in the A-NT group, and extraction of amyloid deposits for proteomics was not feasible. We plan to extend the induction time in a future study to confirm the changes in the proteomics of amyloid deposits as a result of curcumin supplementation.